# Development of Drug-Induced Gene Expression Ranking Analysis (DIGERA) and Its Application to Virtual Screening for Poly (ADP-Ribose) Polymerase 1 Inhibitor

**DOI:** 10.3390/ijms26010224

**Published:** 2024-12-30

**Authors:** Hyein Cho, Kyoung Tai No, Hocheol Lim

**Affiliations:** 1The Interdisciplinary Graduate Program in Integrative Biotechnology & Translational Medicine, Yonsei University, Incheon 21983, Republic of Korea; 2Bioinformatics and Molecular Design Research Center (BMDRC), Incheon 21983, Republic of Korea; 3Baobab AiBIO Co., Ltd., Incheon 21983, Republic of Korea

**Keywords:** drug-induced gene expression, machine learning, ensemble learning, virtual screening, poly (ADP-ribose) polymerase 1

## Abstract

Understanding drug-target interactions is crucial for identifying novel lead compounds, enhancing efficacy, and reducing toxicity. Phenotype-based approaches, like analyzing drug-induced gene expression changes, have shown effectiveness in drug discovery and precision medicine. However, experimentally determining gene expression for all relevant chemicals is impractical, limiting large-scale gene expression-based screening. In this study, we developed DIGERA (Drug-Induced Gene Expression Ranking Analysis), a Lasso-based ensemble framework utilizing LINCS L1000 data to predict drug-induced gene expression rankings. We created novel numerical features for chemicals, cell lines, and experimental conditions, allowing the prediction of gene expression rankings across eight key cell lines. DIGERA outperformed baseline models in the F1@K metric, demonstrating improved precision in gene expression ranking. We also combined DIGERA with an iterative fine-tuning process for de novo design, suggesting 10 PARP1 inhibitors with favorable predicted properties like binding affinity, synthetic accessibility, solubility, membrane permeability, drug-likeness, and similar gene expression ranking to olaparib. Notably, nine compounds were novel, and six analogs of these compounds had references linked to PARP1 inhibition. These results underscore DIGERA’s potential to boost model performance and robustness through novel features and ensemble learning, aiding virtual screening for new PARP1 inhibitors.

## 1. Introduction

Identifying drug-target interactions is crucial for drug discovery, including finding new compounds, understanding side effects, and repurposing approved drugs [1]. Traditional experimental methods for detecting these interactions are costly and time-consuming, creating a significant bottleneck in the drug discovery process. To address this, various computational methods have been developed to efficiently narrow down potential drug-target pairs. These methods include structure-based, ligand-based, and chemogenomic approaches. Structure-based methods rely on the three-dimensional structures of protein targets and often use molecular docking to predict interactions. Ligand-based approaches use experimental data on the chemical properties of compounds and quantitative structure-activity relationship (QSAR) models to predict the activity of new compounds. Chemogenomic methods combine chemical and genomic data to comprehensively map potential drug-target interactions comprehensively, using drug-perturbed gene expression patterns to identify interactions that may be overlooked by other methods. Analyzing gene expression changes following drug treatment can indicate affected proteins, helping identify drugs that influence multiple targets and predict adverse reactions from unintended target interactions [2].

The Library of Integrated Network-Based Cellular Signatures (LINCS) provides over 1.4 million gene expression profiles through the L1000 assay, which measures the expression of 978 landmark genes in response to chemical, genetic, and disease perturbations in a manner of high-throughput profiling [3]. Although the LINCS L1000 datasets provide gene expression profiles perturbed by various chemicals, they contain noise and missing values. Several computational methods have been developed to enhance gene signature detection and predict gene expression profiles of new compounds [2,4]. For example, Woo et al. developed DeepCOP, multilayer perceptrons using binary classification to predict drug effects on gene expression [5]. Similarly, Pham et al. introduced DeepCE, which employs graph neural networks with multi-head attention to capture chemical-gene and gene-gene associations, applying it to COVID-19 drug repurposing [6]. Additionally, Pham et al. also developed CIGER, a method that uses graph convolutional networks (GCN) to predict gene expression profiles induced by chemical structures and rank genes for phenotype-based drug repurposing in pancreatic cancer [2]. These methods enable phenotype-based screening and identification of novel compounds by comparing gene rankings in chemical-induced profiles to those in disease states.

Discovering new molecules with desired properties is challenging due to the vast chemical space, which is estimated to contain about 10^60^ drug-like compounds [7,8]. De novo molecular design creates entirely new molecules with desired characteristics, bypassing existing compounds. Generative deep learning has revolutionized this field, by using input data such as Simplified Molecular-Input Line-Entry System (SMILES) [9] to generate new molecules. The integration of natural language processing with long-short-term memory (LSTM) networks and genetic algorithms (GA) has been used to create new SMILES strings by identifying better molecules than those in the input database through the GuacaMol benchmark [10,11]. Recently, scoring-assisted generative exploration (SAGE) was developed to design chemical inhibitors with desired properties for six protein targets, enhancing generative models through virtual synthesis and specific ring operators while leveraging various QSAR models [12,13,14]. These advancements facilitate efficient exploration of chemical space and the discovery of novel compounds to treat complex diseases like cancer.

Poly (ADP-ribose) polymerase 1 (PARP1) is a 116 kDa enzyme that is essential for DNA repair [15] and plays a pivotal role in cellular survival. PARP1 inhibitors can induce synthetic lethality in cells lacking in homologous recombination capabilities and have been approved for the treatment of several cancers with these deficiencies [16]. Preclinical studies show that PARP1 inhibition enhances radiosensitivity in cancer cells [17], and clinical trials confirm the effectiveness of combining PARP1 inhibitors with radiation therapy [16]. Initially, PARP1 inhibitors were known to cause synthetic lethality by inducing double-strand DNA breaks during replication fork collapse [18]. However, current understanding reveals that PARP1 also modifies chromatin structures through poly ADP-ribosylating histones and other nuclear proteins, resulting in increased genomic instability [16].

In this study, we developed the Drug-Induced Gene Expression Ranking Analysis (DIGERA), a novel computational framework designed to predict gene expression rankings utilizing the LINCS L1000 dataset. We preprocessed the LINCS L1000 Phase II dataset by removing duplicates and clustering gene expression profiles across eight cell lines. We generated numerical features for chemical structures using molecular graphs, fingerprints, and canonical SMILES. For cell lines, we incorporated gene effects from CRISPR/Cas9 and gene ontology; and for experimental conditions through perturbation time and dosage. We constructed machine learning models including Graph Convolutional Networks (GCN), graph-based Transformer, Transformer Encoder, LSTM, and Random Forest Regressors (RFR), evaluating their performance against the baseline (CIGER [2]). We enhanced the models’ performance with ensemble techniques, leading to the refinement of DIGERA. Furthermore, we applied DIGERA in virtual screening for PARP1 inhibitors through a de novo design with SAGE. Consequently, DIGERA is expected to advance drug discovery by predicting gene expression rankings of novel compounds as a part of the chemogenomic screening approach.

## 2. Results

### 2.1. Drug-Induced Gene Expression Ranking Analysis (DIGERA)

To develop the Drug-Induced Gene Expression Ranking Analysis (DIGERA), we preprocessed the LINCS L1000 dataset (Figure 1), which includes gene expression profiles from over a million samples treated with more than 50,000 perturbagens across 98 cell lines [19]. Due to its inherent experimental bias, Qiu et al. introduced a Bayesian-based methodology to extract definitive chemical signatures from the LINCS L1000 Phase II datasets [4]. For our analysis, we used the level 5 dataset curated with these Bayesian methods and selected gene expression profiles from eight core cell lines (A375, A549, HA1E, HEPG2, HT29, MCF7, PC3, and VCAP) at two concentrations (5 and 10 μM) and exposure times (6 and 24 h) to provide a broad spectrum of cellular backgrounds widely used in biomedical research.

To preprocess the dataset, we removed profiles with chemicals having molecular weights over 800 or invalid SMILES strings. We addressed redundant experiments—those with the same chemical and cell line under identical conditions—by comparing the positions of each of the 978 landmark genes across duplicates and selecting profiles that showed the highest frequency of genes near the median value. To further reduce noise, we used k-means clustering with structural fingerprints (MACCS) and gene expression ranking, identifying and removing outlier groups. This process resulted in 15,792 profiles as summarized in Table 1.

### 2.2. Generation of Numerical Features in DIGERA

We developed numerical features for chemicals, cell lines, and experimental conditions to aid machine-learning models in ranking gene expression. We used three molecular descriptors for chemical representation: molecular graphs, molecular fingerprints, and canonical SMILES. The molecular graph captures the two-dimensional connectivity of atoms within a molecule using a graph convolutional network. Molecular fingerprints represent the two-dimensional spatial distribution of atoms, detecting structural similarities with descriptors such as MACCS, PCFP, ECFP, FCFP, and MFBERT. Canonical SMILES provides a text-based representation, converted into binary vectors using one-hot encoding to manage molecular complexity. Each method accurately depicts different aspects of chemical structures, facilitating comprehensive analysis.

Molecular graphs capture the two-dimensional structures of chemicals, detailing atom characteristics (symbol, number of bonded atoms, valence, number of hydrogen atoms, and aromaticity) as well as bond characteristics (type, conjugation, and ring presence). We used various molecular fingerprints: database-based (MACCS, PCFP), connectivity-based (ECFP6, FCFP4), and deep learning-based pre-trained methods (MFBERT) for their robustness in capturing molecular complexity. For canonical SMILES, we converted strings into one-hot encoded vectors using a SMILES dictionary and applied zero padding for standardization. For cell line features, we integrated CRISPR/Cas9 gene knockout data and gene ontology annotations. The CRISPR/Cas9 data provided insights into cell viability changes from knockouts of each of the 978 landmark genes, assessing their functional impact on cell survival. Gene ontology annotations transformed the biological processes associated with each gene into vectors for quantitative analysis, capturing essential genetic information. We summarized this data using PCA to retain crucial genetic information for modeling. Experimental condition features were defined by one-hot encoding perturbation times (6 and 24 h) and chemical dosages (5 and 10 μM), quantifying these variables to enhance understanding of their impacts on gene expression.

Using numerical features, we constructed machine learning algorithms to predict gene expression rankings: GCN (CIGER), graph-based Transformers (Graph Transformer), SMILES-based Transformer Encoder (SMILES Transformer), SMILES-based LSTM (SMILES LSTM), and RFR. We established a baseline model, CIGER, which has been shown to significantly outperform existing models like Random, TT-WOPT, and DeepCOP in prior research [2].

The numerical features varied slightly across models to optimize their capabilities. The RFR model integrates a combination of molecular fingerprints, including MACCS, ECFP6, FCFP4, PCFP, and MFBERT. The SMILES Transformer and SMILES LSTM models use a SMILES encoding method for processing chemical structure information. The CIGER and Graph Transformer models employ a graph-based neural network approach to handle the structural data of molecules. To incorporate experimental conditions and cell line data, the SMILES Transformer, LSTM, and RFR models concatenate molecular fingerprints with numerical features representing CRISPR/Cas9 gene effects and one-hot vectors of experimental conditions. The Graph Transformer combines graph fingerprints with gene ontology descriptors and one-hot vectors for cell line data, exposure time, and drug dosage, ensuring comprehensive data integration to enhance predictive accuracy.

### 2.3. Model Evaluation in DIGERA

To evaluate and compare model performance, we used the F1@K metric, analyzing the ranking of 978 landmark genes into 11 categories: up-regulated (top-1, -10, -50, -100, -200), down-regulated (bottom-1, -10, -50, -100, -200), and others. We did not use the Z-scores from the level 5 dataset directly, as they measure deviations from a gene-specific mean, making direct comparisons between genes’ absolute Z-scores inappropriate due to varying means. Instead, we predicted the relative rankings of gene expressions based on their Z-scores, categorizing them into 11 tiers: up-regulated, down-regulated, and other expressions. This ranking system provides nuanced insights into gene behavior affected by specific perturbagens. Each model underwent training with distinct features and architecture and was fine-tuned through a hyperparameter optimization process (details in Appendix A). The resulting optimized models are summarized in Appendix A.

We developed gene expression ranking prediction models for eight individual cell lines and evaluated their ability to predict up-regulated, down-regulated, and combined up/down-regulated gene rankings. The model performances are detailed in Table 2 and Appendix A, where the heatmaps for the performance in up/down-regulated gene rankings are illustrated in Appendix A. Among the cell lines, models trained on PC3, MCF7, and VCAP showed superior performance.

### 2.4. Model Performance Comparison in DIGERA for Single Cell Lines

In the PC3 cell line, the Voting-based ensemble model had the best results for up/down-regulated genes, with F1-scores of 0.6322 at F1@1, 0.5893 at F1@10, 0.5379 at F1@50, 0.4969 at F1@100, and 0.4330 at F1@200. The Lasso-based ensemble model closely followed with F1-scores of 0.6247 at F1@1, 0.5861 at F1@10, 0.5328 at F1@50, 0.4918 at F1@100, and 0.4289 at F1@200. For up-regulated genes, the Voting-based model achieved F1-scores of 0.6033 at F1@1, 0.5449 at F1@10, 0.4866 at F1@50, 0.4473 at F1@100, and 0.3962 at F1@200. For down-regulated gene predictions, the RFR model excelled at shorter gene rankings, with F1-scores of 0.6616 at F1@1 and 0.6340 at F1@10. The Lasso-based ensemble model performed best at longer gene rankings, with F1-scores of 0.5900 at F1@50, 0.5478 at F1@100, and 0.4718 at F1@200, highlighting variability in model performance across different metrics.

MCF7 was the second best-performing cell line. In predicting combined up/down-regulated gene rankings, the Voting-based ensemble model achieved the highest F1-scores of 0.6040 at F1@1, 0.5681 at F1@10, 0.5173 at F1@50, 0.4789 at F1@100, and 0.4195 at F1@200. For up-regulated genes, this model also recorded the best scores with F1-scores of 0.5772 at F1@1, 0.5294 at F1@10, 0.4727 at F1@50, 0.4341 at F1@100, and 0.3863 at F1@200. The Lasso-based ensemble model led in predicting down-regulated gene rankings, achieving F1-scores of 0.6222 at F1@1, 0.6057 at F1@10, 0.5619 at F1@50, 0.5243 at F1@100, and 0.4541 at F1@200.

In the VCAP cell line, models showed variability in performance across different metrics for up/down-regulated gene rankings. The RFR model showed the highest F1@1 of 0.6051 at shorter gene rankings. The Lasso-based ensemble model achieved F1-scores of 0.5791 at F1@10 and 0.5344 at F1@50. The Voting-based ensemble model performed best at longer gene rankings, with F1-scores of 0.5014 at F1@100 and 0.4479 at F1@200. For up-regulated genes, the RFR model excelled at shorter gene rankings, with F1-scores of 0.6323 at F1@1 and 0.5938 at F1@10, while the Voting-based ensemble model showed the best performance at longer gene rankings, recording F1-scores of 0.5313 at F1@50, 0.4924 at F1@100, and 0.4372 at F1@200. For down-regulated genes, the RFR model performed best at shorter gene rankings, achieving 0.5753 at F1@1, while the Lasso-based ensemble model performed best at longer gene rankings, recording F1-scores of 0.5686 at F1@10, 0.5400 at F1@50, 0.5118 at F1@100, and 0.4592 at F1@200.

In the other five cell lines, the RFR and two ensemble models (Voting-based and Lasso-based) generally outperformed other models. For up/down-regulated genes, the RFR showed excellent performance across all metrics from F1@1 to F1@200 in the HT29 cell line. The Lasso-based ensemble model excelled from F1@1 to F1@200 in the A549 and HA1E cell lines and up to F1@100 in the A375 and HEPG2 cell lines. The Voting-based ensemble model demonstrated its strength at F1@200 in the A375 and HEPG2 cell lines. For predicting up-regulated genes, both the RFR and Lasso-based ensemble models performed well at shorter gene rankings, while the Voting-based ensemble model excelled at longer gene rankings. In predicting down-regulated genes, the Lasso-based ensemble model consistently showed the best performance, whereas the RFR was particularly effective at F1@200 in the HA1E and HT29 cell lines.

We compared the performance of nine machine learning models trained on eight single cell lines, as illustrated in Figure 2A. The Lasso-based ensemble, RFR, and Voting-based ensemble models demonstrated top-tier performance, outshining the other six models. These three models alternated in their high rankings, followed by the Graph Transformer and CIGER (baseline), which utilized two-dimensional structural data. Models employing text-based SMILES data, such as the SMILES LSTM and SMILES Transformer, ranked next. Ensemble models based on Ridge or ElasticNet showed lesser performance compared to single-model approaches.

In a comparative analysis of the eight cell lines shown in Figure 2B, PC3 displayed the strongest performance in down-regulation prediction, significantly outperforming both MCF7 and VCAP, although MCF7 also outperformed VCAP in this measure. VCAP excelled in prediction for up-regulated and up/down-regulated genes, surpassing PC3 and achieving the highest scores in up-regulated gene predictions. Beyond the top performers, HT29 ranked fourth, while A375 and HEPG2 tied for fifth and sixth, each showing strengths at different gene ranking levels: A375 excelled at longer levels (F1@1 and F1@10), and HEPG2 performed better at shorter levels (F1@50, F1@100, and F1@200). A549 showed the lowest performance. Interestingly, HT29 led in up-regulated gene predictions, with A549 close behind, indicating a robust capability in up-regulated genes despite weaker performance in up/down-regulated genes. HEPG2 and A375 excelled in specific aspects of down-regulated gene predictions, with HEPG2 excelling in shorter rankings and A375 in longer rankings. These results highlight variability across cell lines, with distinct strengths and weaknesses in predicting up or down-regulations, while HA1E, a normal cell line, demonstrated moderate capabilities across all metrics, typical of the subdued responses of healthy cell lines to drug stimuli.

### 2.5. Model Performance Comparison in DIGERA for Multiple Cell Lines

To further explore the predictive abilities of various cell lines, we constructed models using two datasets: one that combined three high-performing cell lines (MCF7, PC3, and VCAP) and another encompassing all eight cell lines, which consisted of five additional lower-performing cell lines (A375, A549, HEPG2, HT29, and HA1E). The performance outcomes of these models are detailed in Table 3 and Appendix A, with the performance of the best model (Lasso-based ensemble model) depicted in Figure 2C.

Within the dataset featuring the three cell lines (MCF7, PC3, and VCAP), the Lasso-based ensemble model demonstrated exceptional performance, achieving the highest F1-score of 0.6074 at F1@1. Conversely, the Voting-based ensemble model excelled across broader rankings, with F1-scores of 0.5776 at F1@10, 0.5301 at F1@50, 0.4930 at F1@100, and 0.4343 at F1@200. In predicting up-regulated genes, the Voting-based ensemble model emerged as the leading performer, recording an F1-score of 0.6012 at F1@1 and following with scores of 0.5567 at F1@10, 0.4995 at F1@50, 0.4605 at F1@100, and 0.4084 at F1@200. For down-regulated genes, the Lasso-based ensemble model consistently outperformed other models, achieving its highest scores with 0.5985 at F1@10, 0.5612 at F1@50, 0.5264 at F1@100, and 0.4612 at F1@200. Additionally, the RFR model showed the highest F1-score of 0.6150 at F1@1 for the shortest gene rankings.

The combined dataset from the three cell lines (MCF7, PC3, and VCAP) demonstrated higher scores compared to the top-performing individual cell line model in predicting up/down-regulated genes. It achieved scores comparable to the individual models of MCF7, VCAP, and PC3, as shown in Figure 2C. Specifically, the three-cell line model reached F1-scores of 0.6074 at F1@1, 0.5776 at F1@10, 0.5301 at F1@50, 0.4930 at F1@100, and 0.4343 at F1@200, slightly underperforming the PC3 model, which recorded 0.6322 at F1@1, 0.5893 at F1@10, 0.5379 at F1@50, 0.4969 at F1@100, and 0.4330 at F1@200. However, it outperformed MCF7 across all metrics and surpassed VCAP in longer-ranking genes. Contrasting trends emerged between the combined three-cell line model and the individual top-performing cell lines in predicting up or down-regulated genes. The combined model showed advantages in up-regulation, except for the F1@1 of PC3, which was 0.6033. The model scored 0.6012 at F1@1, 0.5567 at F1@10, 0.4995 at F1@50, 0.4605 at F1@100, and 0.4084 at F1@200. Furthermore, it demonstrated superior down-regulation performance compared to VCAP across all metrics and outperformed MCF7 in longer-ranking predictions, achieving 0.6150 at F1@1, 0.5985 at F1@10, 0.5612 at F1@50, 0.5264 at F1@100, and 0.4612 at F1@200. The model with three cell lines showcased robust predictive capabilities in up- or down-regulated gene prediction.

In the comprehensive dataset containing all eight cell lines, the Voting-based ensemble model was the leading performer in up/down-regulated gene prediction, achieving F1-scores of 0.5628 at F1@1, 0.5264 at F1@10, 0.4781 at F1@50, 0.4437 at F1@100, and 0.3930 at F1@200. It also excelled in up-regulated gene prediction, recording F1-scores of 0.5768 at F1@1, 0.5276 at F1@10, 0.4693 at F1@50, 0.4323 at F1@100, and 0.3832 at F1@200. However, in down-regulated gene prediction, the Lasso-based ensemble model demonstrated superior performance, achieving F1-scores of 0.5499 at F1@1, 0.5261 at F1@10, 0.4885 at F1@50, 0.4568 at F1@100, and 0.4033 at F1@200. We also evaluated the performance of a dataset comprising all eight cell lines against a dataset containing the top three cell lines. It revealed that the models trained with all eight cell lines did not outperform those with the three cell lines and failed to exceed the performance of the individual models of MCF7, PC3, and VCAP across all metrics. Conversely, the models of all eight cell lines outperformed the lower-tier five-cell model (A549, A375, HEPG2, HT29, and HA1E) in up-regulated gene prediction, whereas they exhibited weaknesses in predicting down-regulated genes.

In a comparative analysis of multiple cell line combinations, both the Voting-based and Lasso-based ensemble models exhibited competitive performance across all metrics. The Voting-based model excelled in predicting up/down-regulated and up-regulated gene predictions, while the Lasso-based model showed strengths in down-regulated gene prediction. Additionally, the RFR model demonstrated intermediate predictive power across all metrics. Assessing the performance of combinations using the top three cell lines revealed stronger overall performance compared to individual cell lines, with the notable exception of PC3. This suggests that combining datasets has the potential to exceed the performance of single cell line models, although it did not surpass PC3. Despite this, the three-cell line combination outperformed MCF7 in all metrics and VCAP in longer gene rankings. However, the comprehensive dataset of all eight cell lines did not outperform the three-cell line dataset and showed weaknesses in predicting down-regulated genes. Nonetheless, the dataset of all eight cell lines still demonstrated notably better performance compared to the lower-tier five-cell lines. Therefore, we chose the Lasso-based ensemble model trained with the top three cell lines as the DIGERA model for effectively predicting up/down-regulated gene rankings.

### 2.6. Application of DIGERA to De Novo Design for PARP1 Inhibitors

To identify novel compounds that target PARP1, we combined the DIGERA model with SAGE to perform a de novo design. First, the DIGERA model was employed to predict the gene expression ranking of compounds by measuring the Spearman correlation between the rankings of 978 landmark genes and the experimental gene rankings observed with olaparib at a concentration of 10 μM in MCF7 cells over 24 h. For comparison, we evaluated the Spearman correlation between the experimental gene rankings under the same conditions and the DIGERA-predicted rankings for two known PARP1 inhibitors, olaparib [20] and AG14361 [21,22], which demonstrated correlation scores of 0.5520 and 0.4609, respectively. Secondly, we developed a classification model capable of predicting the target specificity of PARP1. This was achieved by compiling a dataset of compounds known to specifically target PARP1, including active, inactive, and decoy compounds, as detailed in Table 1. Our goal was to develop a classification model capable of differentiating active compounds, defined as those with a binding affinity stronger than 1 μM, from inactive or decoy compounds. The model training involved hyperparameter optimization, as outlined in Appendix A, along with 10-fold cross-validation. The models with the best hyperparameters are listed in Appendix A, and their performance metrics are detailed in Table 4. The most effective model, an LGBM classifier using MACCS/ECFP6 features, achieved an F1-score of 0.974 in the training set, 0.937 in the validation set, and 0.956 in the test set. The accuracy, precision, recall, and MCC of the optimally tuned models are summarized in Appendix A, where the top-performing model (MACCS/ECFP/LGBM) demonstrated an accuracy of 0.992, a precision of 0.927, a recall of 0.986, and an MCC of 0.952 in the test set.

Next, we optimized the properties for target specificity using the SAGE method [12], where the best-performing classification model was integrated into SAGE for 50 iterative steps of fine-tuning aimed at designing PARP1 inhibitors. To restrict our chemical exploration to drug-like molecules, we applied Muegge’s drug-likeness criteria as chemical filters. We incrementally incorporated scoring metrics for target specificity (Score 1), synthetic accessibility and solubility (Score 2), membrane permeability (Score 3), drug-likeness (Score 4), and gene expression ranking (Score 5), with the results illustrated in Figure 3. Using SAGE over 50 steps, we achieved median scores above 0.75 by the 4th step and above 0.90 by the 5th step for Score 1. For Score 2, the median score surpassed 0.75 by the 3rd step and 0.90 by the 5th step. For Score 3, the system identified molecules scoring over 0.75 by the 2nd step and above 0.90 by the 5th step. For Score 4, the median score crossed the 0.75 threshold by the 3rd step and reached above 0.90 by the 8th step. When optimizing Score 5, SAGE achieved a median score above 0.75 by the 4th step and over 0.85 by the 9th step. However, it did not reach 0.9 within 50 steps. Consequently, we employed the iterative fine-tuning strategy focused on enhancing Score 5, a metric that accounts for target specificity, synthetic accessibility, aqueous solubility, membrane permeability, drug-likeness, and gene expression ranking. Through this approach, we successfully generated 708,755 compounds.

To pinpoint optimal molecules for targeting PARP1, we meticulously filtered all molecules generated by SAGE to maximize Score 5. Initially, we applied a stringent threshold of 0.9 for target specificity, synthetic accessibility, aqueous solubility, and membrane permeability, and a more lenient threshold of 0.5 for drug-likeness and gene expression ranking. Based on these criteria, we selected the top 100 compounds and predicted their binding affinity using molecular docking, ultimately selecting the top 10 hits based on docking scores, as illustrated in Figure 4A. All selected compounds (**1**–**10**) demonstrated favorable docking scores with PARP1, ranging from −13.341 to −14.080. Notably, except for Compound **3**, none of these compounds has a CAS Number, indicating their novelty. When we searched for structurally similar compounds with CAS Numbers, several analogs were found to have references related to PARP inhibition. The **3** is patented as a PARP inhibitor (WO2004-GB1059), and the analog of **7** is also patented as a PARP inhibitor (WO2022017508). The analog of **9** is patented as a PARP1 inhibitor (WO2006021801), and the analog of **10** was identified as a PARP1 inhibitor through a 3D-QSAR study [23]. Additionally, the analogs of **2** and **8** are patented as PARP7 inhibitors (WO2023-IB50507 and WO2023139536, respectively).

Further validation was conducted for the novel compound **1**. To assess the molecular interactions between these compounds and PARP1, molecular docking, dynamics simulations, and quantum-mechanical FMO analysis were performed, as shown in Figure 4B. Subsequent molecular dynamics simulations confirmed that the PARP1 protein-ligand complexes with these compounds maintained stable binding, with fluctuations within 1–3 Å of the thermal average throughout the simulation period. Based on the quantum-mechanical FMO analysis results, compound **1** interacted with 11 common residues: D766, H862, G863, S864, R878, I879, Y896, A898, S904, Y907, and E988. Thus, our results confirm that the DIGERA model can be effectively utilized alongside de novo design for virtual screening to reliably rank compounds.

## 3. Discussion

Understanding drug-target interactions at the molecular level is crucial for optimizing therapeutic agents. Gene expression profiling, which analyzes alterations in gene expression following drug exposure, has become an indispensable tool in this regard. It helps identify potential interactions and predict a drug’s behavior in the body, which is essential for evaluating its efficacy and safety. By assessing how a drug modulates gene expression and comparing these patterns against established profiles, gene expression-based screening can preemptively eliminate chemicals with undesirable side effects and infer shared mechanisms of action among drugs. Here, we propose DIGERA, a robust drug-induced gene expression ranking analysis using ensemble learning for novel compounds. Our model outperformed the baseline (CIGER) and demonstrated improved performance compared to single models. Notably, DIGERA extends existing frameworks by broadening the feature space and integrating ensemble techniques, resulting in performance gains in F1@K metrics. Furthermore, the incorporation of generative models, such as SAGE, facilitated de novo molecular design, enabling the suggestion of PARP1 inhibitors with desirable physicochemical properties and novel chemical structures. However, DIGERA’s computational complexity and extensive preprocessing requirements present challenges for scalability. Future work will focus on optimizing computational efficiency and experimentally validating the biological relevance of the identified compounds to confirm their gene expression patterns and therapeutic potential.

The DIGERA predicted gene expression rankings rather than the exact values of differentially expressed gene signatures in response to drug-induced perturbations. Traditionally, differentially expressed genes have been characterized by their relative frequencies within gene sets. Gene expression rankings offer enhanced robustness against technical variability across experiments and batches compared to absolute expression values, which may not adequately represent the underlying biological mechanisms of drug-induced perturbations. To address this, we utilized the relative order of expression among the 978 genes for ordinal regression. The principles of ordinal regression, along with its associated loss functions, align with the task of ordering expressed signatures or categorical items, employing penalties based on rankings. This approach provides a more nuanced and robust analysis of drug-perturbed gene expression patterns, potentially yielding more biologically meaningful insights [24]. While this method improves the analysis of gene expression, we further explored the predictive power of various ensemble models to enhance performance. Lasso-based ensemble models outperformed both the baseline and individual predictive models by offsetting individual model errors through averaging and L1 regularization. However, not all ensemble strategies yielded performance improvements; for example, Ridge and ElasticNet ensembles showed a decline in performance, likely due to the L2 penalty’s heightened sensitivity to outliers, which exacerbated errors inherent in individual models.

Moreover, our gene expression ranking prediction model remains constrained by several issues linked to the dataset employed. The LINCS L1000 dataset, crucial for both training and evaluation, is marred by numerous missing expression values and significant noise across diverse chemicals, concentrations, and cell lines. Prior studies by Lim and Pavlidis have criticized the dataset for its weak correlation with earlier benchmarks like the CMap1 dataset and noted poor reproducibility issues [25]. Although Qiu et al. have proposed a Bayesian-based peak deconvolution algorithm to mitigate some of these challenges, offering unbiased likelihood estimations for peak locations and z-score-based peak characterization [4], we observed that discrepancies in gene signature values persist. These inconsistencies are prevalent among different studies utilizing the LINCS L1000 data, underscoring the persistent influence of experimental conditions on data quality. To mitigate these dataset limitations, our approach incorporated the use of preprocessed gene signatures and targeted selection of specific cell line clusters to reduce noise effects. However, this strategy inevitably constrained the quantity of data available for constructing our models. Despite these obstacles, ongoing efforts by numerous researchers to refine this dataset through various statistical methods or transfer learning indicate a collective endeavor to develop more robust predictive models [2,4,6,26].

The application of advanced machine learning techniques to the L1000 dataset will significantly enhance predictive models, accelerating drug discovery. These advancements will enable the early identification of off-target effects and adverse reactions, improving the accuracy of predictions for compound interactions across various cell lines. Such capabilities will mitigate risks associated with drug development and prevent costly failures. Furthermore, enhanced models will refine drug targeting and design by deepening our understanding of molecular mechanisms within biological systems. This improvement will streamline the virtual screening process, as predictive models will better identify compounds likely to exhibit biological activity. Researchers will then be able to focus on the most promising candidates for detailed analysis, saving time and resources while enhancing the throughput and efficiency of the screening process. This will facilitate a quicker transition from compound synthesis to clinical testing. While DIGERA has been designed to maximize the utility of LINCS L1000 data, integrating additional omics layers, such as proteomics, metabolomics, or transcriptomics, could further enhance its predictive power by providing a more comprehensive view of cellular responses. The modular nature of DIGERA makes it well-suited for future expansions to incorporate multi-omics datasets as they become increasingly available. Moreover, real-time feedback mechanisms, such as dynamically adjusting model parameters based on experimental outcomes, represent a promising direction for future developments. These enhancements could enable the iterative refinement of drug candidates in silico, aligning computational predictions more closely with experimental and clinical results. Ultimately, these machine learning models will not only bolster the robustness of the L1000 dataset but also transform the broader fields of pharmacology and toxicology, advancing our capabilities to discover and optimize novel therapeutics through more precise and predictive methodologies.

## 4. Materials and Methods

### 4.1. The LINCS L1000 Gene Expression Datasets in DIGERA

The LINCS datasets provide a comprehensive repository of cellular response patterns to various perturbagens [3]. The L1000 assays offer a high-throughput solution for gene expression analysis of 978 landmark genes, profiling over a million samples treated with more than 50,000 perturbagens across 98 cell lines [19]. Qiu et al. developed a Bayesian-based pipeline to extract signatures from the LINCS L1000 Phase II datasets [4]. We used the Bayesian-based curated level 5 dataset containing differential gene expression signatures and selected profiles from the eight largest cell lines (A375, A549, HA1E, HEPG2, HT29, MCF7, PC3, and VCAP) at concentrations of 5 and 10 μM, with exposure times of 6 and 24 h. The selected cell lines include A375 (melanoma), A549 (lung carcinoma), HA1E (immortalized kidney epithelial), HEPG2 (hepatocellular carcinoma), HT29 (colorectal adenocarcinoma), MCF7 (breast carcinoma), PC3 (prostate adenocarcinoma), and VCAP (vertebral metastasis of prostate adenocarcinoma). This diverse selection encompasses various cancer types and one immortalized normal cell line, allowing for a comprehensive examination of cellular responses to the treatment conditions.

For redundant experiments using the same chemical and cell line under identical conditions, we compared the position of each L1000 gene within these duplicates and selected profiles showing the highest frequency of genes approximating the median value.

The level 5 dataset uses Z-scores to analyze gene expression deviations, indicating how many standard deviations a gene’s expression is from its mean. A positive Z-score means a gene’s expression is above the mean, while a negative Z-score indicates it is below. To handle varying distributions and expression magnitudes, we transitioned from predicting absolute Z-scores to predicting relative rankings based on Z-scores. We converted Z-scores into ranked categories based on their deviation from the median, classifying gene expressions into 11 tiers: the highest (up-regulated) levels of 1, 10, 50, 100, and 200, and the lowest (down-regulated) levels of 1, 10, 50, 100, and 200, plus an additional category for all other expressions. Genes with identical Z-scores were assigned the same average rank to reduce the inherent noises in the LINCS platform. The aligned 978 gene rank presents that the order of expression among gene sets is more critical than the absolute expression values or precise relative frequencies.

### 4.2. Numerical Features in DIGERA

To generate numerical features for chemical structures, we employed three distinct methodologies: molecular graphs, molecular fingerprints, and canonical SMILES. Molecular graphs represent the two-dimensional structures of chemicals, capturing detailed atom characteristics such as symbol, number of bonded atoms, valence, number of hydrogen atoms, and aromaticity, as well as bond specifics like type, conjugation, and presence in a ring. We utilized a variety of molecular fingerprints, including database-based (MACCS, PCFP), connectivity-based (ECFP6, FCFP4), and deep learning-based pre-trained (MFBERT) fingerprints, selected for their efficacy in depicting molecular complexity. For canonical SMILES, we transformed the strings into one-hot encoded vectors using a specific SMILES dictionary and applied zero padding to achieve uniform lengths across all molecules.

Firstly, we utilized a molecular graph through the application of a graph convolutional network (GCN) [27], as described by Pham et al. [2]. The GCN framework consists of atoms represented as nodes and chemical bonds depicted as edges. Our model incorporated a two-layer GCN with a radius of 2, which computes a 1024-dimensional fingerprint for each molecule. This network updates node attributes using aggregated information from neighboring nodes. For atom nodes, it uses several one-hot encoded vectors capturing chemical characteristics: atom symbols (44-dimensional), degree (6-dimensional), hydrogen atoms (5-dimensional), valence (6-dimensional), and aromaticity (2-dimensional). Bond edges are similarly represented by vectors indicating bond type (4-dimensional for single, double, triple, and aromatic), conjugation (2-dimensional), and ring belonging (2-dimensional). The GCN outputs nodes that encapsulate a molecule’s sub-structure, highlighting essential properties such as atom symbol, connectivity, and bond type.

Secondly, we incorporated pre-defined chemical fingerprints for scrutinizing compounds within extensive large databases. The Molecular Access System (MACCS) is a common fingerprint for assessing structural similarity using 166-bit MACCS keys [28], while the PubChem system employs substructural fingerprints (PCFP) with 881-bit structural keys to embody chemical structures, thereby facilitating similarity and neighbor searches [29]. In addition, we concatenated fingerprints derived from the atom connectivity within the compounds. The Extended-Connectivity Fingerprints (ECFP) were designed for structure-activity relationship modeling and signifying circular atom neighborhoods [30], while Function-Class Fingerprints (FCFP) uniquely index the pharmacophore-like roles of distinct atoms within the molecules. We created ECFP with a diameter of 6 (ECFP6) and FCFP with a diameter of 4 (FCFP4) as 1024-bit keys using RDKit version 2021.9.5.1 [31] and Morgan algorithms [32]. Furthermore, we considered pre-trained chemical embeddings based on natural language processing (NLP) methods, which are adept at comprehending contextual relationships in a sequence. Chemical structures consist of intricate patterns, and NLP models, by leveraging their ability to capture complex relationships, can be employed to create chemical embedding features. The Molecular Fingerprints through Bidirectional Encoder Representations from Transformers (MFBERT) is a pre-trained transformer model with a consolidated dataset of over 1.2 billion molecules [33]. For the prediction model of gene expression, we utilized the most complex fingerprints by concatenating all available fingerprints.

Lastly, SMILES representations were transformed into binary vectors through one-hot encoding. To ensure data uniformity, chemicals with SMILES exceeding 120 tokens and molecular weights greater than 600 g/mol were excluded. Each character in the SMILES string, representing a distinct chemical group, was mapped to a predefined dictionary of 47 characters, which included special tokens for start, end, and padding. The binary vectors were then constructed by concatenating these character vectors. To accommodate variations in SMILES lengths, padding was implemented in the one-hot encoding process, adding zeros to each sequence to achieve a uniform maximum length of 120.

To create numerical features for cell lines, we collected 17,453 gene knockout effects across 1078 cell lines from the CRISPR knockout screening data in the Cancer Dependency Map (DepMap 22Q4) [34]. These effects elucidate relationships between cell lines, as specific gene knockouts can affect cellular behavior. We performed principal component analysis (PCA) to reduce dimensionality while preserving important variance, selecting 152 principal components for the eight cell lines, accounting for about 50.1% of variances. For experimental conditions, we used one-hot vectors for compound concentration and exposure time.

### 4.3. Machine Learning in DIGERA

To create prediction models for gene expression ranking of the 978 landmark genes, we used algorithms such as GCN (CIGER [2]), graph-based Transformers (Graph Transformer), SMILES-based Transformer Encoder (SMILES Transformer), SMILES-based LSTM (SMILES LSTM), and RFR. As our baseline model, we implemented the Chemical-Induced Gene Expression Ranking (CIGER) framework [2], which utilizes graph convolutional networks (GCN) and multi-layer feedforward neural networks (FNN) to predict gene expression rankings and classification tasks on the LINCS L1000 dataset. The CIGER framework consists of three key components: GCN, multi-head attention mechanism, and FNN. The GCN serves as the feature mapping component, transforming SMILES strings into graph representations to extract relevant information from the chemical structures. The multi-head attention component of CIGER focuses on the 978 landmark genes from the L1000 dataset, capturing interdependencies among genes, chemicals, and cell lines, thereby generating contextualized representations for each gene. The FNN, consisting of two layers in an encoder-decoder configuration, predicts the ranking of the 978 landmark genes in gene expression profiles. CIGER also utilizes ranking loss functions to emphasize the significance of ranking information within these profiles, enhancing the predictive accuracy and relevance of the results. We re-trained the CIGER model as a base model to predict the ranking of landmark genes with one layer, one multi-head, and a list-wise rank cosine loss function.

To improve the predictive performance of the CIGER framework, we introduced modifications to develop graph-based transformer models (Graph Transformer). Firstly, we increased the number of layers and attention heads in the model’s architecture to enhance its structural complexity. This enhancement is designed to enable a deeper and more nuanced analysis of the complex patterns observed in gene expression data, thereby facilitating a more detailed and accurate interpretation. Secondly, we revised the ranking loss functions to increase the model’s sensitivity to the hierarchical order of gene expressions. This adjustment aims to ensure more precise and reliable predictions by better aligning the model’s output with the underlying biological structures.

A transformer is specifically designed to manage sequential data through NLP tasks, utilizing attention mechanisms [35]. These mechanisms help the model discern the contextual relevance of positions within a sequence and assess the importance of each element. Specifically, the transformer encoder is tasked with learning language patterns by calculating the likelihood of word sequences. In a specialized application, the SMILES Transformer predicts subsequent sequences of SMILES notations. The tokens from these sequences are embedded within the embedding layer. The self-attention layers of the transformer encoder focus on specific positions in the sequence, using masked multi-head attention to preserve the input order. Finally, the output from the transformer encoder is concatenated with numerical features related to cell lines and experimental conditions of time and dosage. This combined data is then processed through a linear layer, culminating in a log-softmax function that generates the final output (gene expression ranking).

Long-short-term memory (LSTM) is a specialized type of recurrent neural network (RNN) designed to address the vanishing gradient problem commonly associated with traditional RNNs [36,37]. The LSTM model incorporates a sophisticated system of gates that manage the flow of information, allowing it to selectively retain or discard data across extended sequences. This feature is particularly useful for processing the complexities of language-like data sequences. We enhanced the LSTM model by integrating positional encoding to effectively capture the positional information of elements within the SMILES sequences, similar to techniques used in Transformer models. This adaptation ensures that the sequential order contributes to the model’s learning process, enhancing its ability to recognize patterns dependent on the sequence of inputs. Furthermore, akin to the approach in the SMILES Transformer, the encoding layer dedicated to cell line data is processed through the LSTM model. This processed data is then augmented with a one-hot encoding of variables such as chemical exposure time and concentration, along with other numerical features pertinent to the study. The combined tensor undergoes further processing in a linear layer to generate the final output. This modified LSTM architecture, which we refer to as the SMILES LSTM, is specifically tailored to handle the intricacies of chemical structure representation in SMILES format. By capturing both the chemical and positional data effectively, the SMILES LSTM offers a robust method for modeling complex interactions in biochemical datasets.

The random forest regressor (RFR) model is a robust ensemble method that utilizes a multitude of decision trees to enhance both stability and precision in predictions. By constructing several decision trees during training and outputting the mean prediction of the individual trees, the RFR model effectively minimizes variance and reduces the risk of overfitting through an averaging process across different trees [38]. To further adapt the RFR model for handling more complex multidimensional data, we have modified it to support a multi-output framework. This enhancement allows the RFR model to manage gene expression profiles, which involve predicting multiple outputs simultaneously. By integrating this multi-output capability, the RFR model can now learn relationships across different genes within a single model run, thereby providing a comprehensive analysis of gene expression profiles based on input features such as chemical structures, cell lines, and experimental conditions.

To find the most effective regression models for gene expression, we conducted a grid search across several models, utilizing specific hyperparameters for each. For the RFR, we adjusted the number of gradient-boosted trees (n_estimators), the maximum tree depth for base learners (max_depth), and the number of features considered during the best-split selection (max_features). In the case of the Graph Transformer, we explored variations in the number of layers, the number of multi-heads, and the types of loss functions (point-wise mean squared error and list-wise rank cosine). List-wise rank cosine loss is related to ranking and recommendation systems that optimize the cosine similarity between the predicted and true rankings of items [24]. For the SMILES Transformer, we adjusted the number of layers, embedding sizes (dmodel), hidden unit sizes, the number of multi-heads, and the types of loss functions (mean squared error and list-wise rank cosine). Similarly, for the SMILES LSTM, we modified the number of layers, embedding sizes (dmodel), hidden unit sizes, and the types of loss functions (mean squared error and list-wise rank cosine). The detailed hyperparameter tuning procedure for these gene expression prediction models is summarized in Appendix A.

Ensemble methods harness the collective power of multiple machine-learning algorithms to improve predictive performance. To make ensemble models, we utilized four regularization methods (Voting, Lasso, Ridge, and ElasticNet) with the individual models (RFR, Graph Transformer, SMILES Transformer, and SMILES LSTM) built with optimal hyperparameters. Voting regression aggregates predictions from multiple models by averaging their outcomes to form a final prediction. Least absolute shrinkage selector operator (Lasso) regularization is a linear model incorporating L1 regularization, and its regularization term penalizes large coefficient sizes, effectively shrinking coefficients towards zero and favoring sparse solutions [39,40]. Similarly, Ridge regularization employs L2 regularization, particularly effective when independent variables are highly correlated. This regularization technique mitigates multicollinearity issues, stabilizing regression coefficients and reducing overfitting risks [41]. Elastic Net (ElasticNet) regularization combines L1 and L2 regularization penalties to shrink predictor coefficients, striking a balance between sparsity and multicollinearity [42].

We evaluated model performance on cross-validation sets using F1@K metrics for gene expression ranking tasks. The F1@K is the harmonic mean of Precision@K and Recall@K ($\frac{2\times \mathrm{Precision}@K\times \mathrm{Recall}@K}{\mathrm{Precision}@K+\mathrm{Recall}@K}$). Precision@K calculates the fraction of correctly predicted genes in the top-K group, while Recall@K measures the proportion of relevant items correctly identified within the top-K predictions. Relevant items were defined as the top 200 genes for up-regulated predictions, the bottom 200 for down-regulated predictions, and both for combined predictions. We assessed performance at various K-levels, including 1, 10, 50, 100, and 200.

### 4.4. PARP1 Classification Models

The ligand structures for PARP1 classification models were obtained from the DUD-E benchmarking sets [43] and ChEMBL32 [44]. Active compounds were defined as those with Ki, Kd, and IC50 values better than 1 μM, while compounds with activity worse than 1 μM or undetermined were classified as inactive. To increase the number of inactive compounds, we used DUD-E decoy protocols [43], generating a decoy pool from the ZINC database [45] based on six properties of inactive compounds prepared at pH 6–8 (molecular weight, LogP, rotatable bonds, hydrogen bond acceptors, hydrogen bond donors, and net charge).

To create QSAR models for PARP1 inhibitors, we used the Random Forest Classifier (RFC), Light Gradient Boosting Machine (LGBM), and Extreme Gradient Boosting (XGB) classifiers. These methodologies utilize decision trees to prevent overfitting and decrease variance. Each decision tree inspects numerical characteristics to produce continuous results, and these trees are constructed sequentially and calibrated to the discrepancies between real and predicted values produced by preceding trees. The hyperparameter tuning procedure for the PARP1 inhibitor classification models is summarized in Appendix A. To identify the most effective classification models for PARP1 inhibitors, we performed a grid search by considering three hyperparameters in LGBM, two in RF, and four in XGB. For LGBM, we utilized the type of boosing methods (boosting_type), the number of gradient-boosted trees (n_estimators), and the boosting learning rate (learning_rate) [46]. For RFC, we used the number of gradient-boosted trees (n_estimators) and the number of features to contemplate during the best-split selection (max_features) [38]. For XGB, we considered the type of boosting methods (booster), the number of gradient-boosted trees (n_estimators), the maximum tree depth for based learners (max_depth), the number of features to consider when looking for the best split (max_features), and the boosting learning rate (learning_rate) [47].

For PARP1 classification models, we used a stratified split approach, with 80% of the data for training and 20% for testing, using a fixed random seed in Scikit-learn [38]. The training set underwent 10-fold cross-validation to optimize hyperparameters with GridSearchCV. We created sophisticated fingerprints (MACCS/PCFP, MACCS/ECFP6, MACCS/FCFP4, and MACCS/MFBERT) by merging MACCS keys with other fingerprints (PCFP, ECFP6, FCFP4, and MFBERT) for the classification models. Details of these models are in Appendix A. To evaluate the performance of the PARP1 inhibitor classification models on the test sets, we used four metrics: Accuracy, Precision, Recall, F1-score, and Matthews correlation coefficients. Accuracy is the proportion of correct predictions out of all predictions. Precision, or positive predictive value, is the ratio of true positives to predicted positives. Recall, or sensitivity, indicates how many actual positives were correctly classified. The F1-score is the harmonic mean of Precision and Recall. The Matthews correlation coefficient (MCC) is a balanced metric for evaluating the accuracy of classification, accounting for true and false positives and negatives.

### 4.5. Scoring-Assisted Generative Exploration (SAGE)

Scoring-assisted generative exploration (SAGE) uses a framework combining autoregressive NLP, chemical diversification operators, and various QSAR/QSPR scoring models to generate high-scoring molecules aligned with specified objectives [12]. We also included five chemical filters to assess molecule quality at each generation stage as follows: rule-based drug-likeness, synthetic accessibility, aqueous solubility, membrane permeability, and overall drug-likeness.

Initially, we applied the Muegge filter [48] for rule-based drug-likeness, categorizing molecules based on their similarity to known drug-like compounds, excluding those with molecular weights over 600 or under 200, LogP values over six, more than six hydrogen donors, over 12 hydrogen acceptors, more than 15 rotatable bonds, more than seven aromatic rings, fewer than two heteroatoms, or less than five carbon atoms. Next, we evaluated synthetic accessibility using the retrosynthetic accessibility score (RAscore) [49], and aqueous solubility using SolTranNet [50]. We also assessed Caco-2 membrane permeability [51] and human intestinal absorption (HIA) [52], essential for estimating drug permeation through intestinal tissues. The prediction models for Caco-2 and HIA were derived from our previous study [12]. Finally, we used an RNN-based model for drug-likeness, trained to learn the probability distribution of known drugs [53]. This model differentiates the drug-likeness of molecules from various sources—randomly generated molecules, commercially available compounds, bioactive molecules, and FDA-approved drugs—by providing distinct drug-likeness distributions for each.

To incorporate the chemical filters and scoring systems into SAGE, we defined one chemical filter and four scoring stages. Once molecules are generated by SAGE, the rule-based drug-likeness chemical filter excludes any invalid or out-of-criteria molecules. For Score 1, we focused solely on target specificity, utilizing the highest prediction score from the PARP1 classification model. For Score 2, we integrated target specificity, synthetic accessibility, and apparent solubility by averaging Score 1, the RAscore, and the solubility score. We assigned a value of 0 if the LogS value was less than −8.0 and a value of 1 if it was greater than −4; values between −4 and −8 were linearly interpolated between 0 and 1. For Score 3, we added the membrane permeability score to Score 2, which was calculated from the average scores of the predictions from the HIA classification model and those from Caco-2 regression model’s predictions. We assigned a value of 0 if the Caco-2 value was below −6.85 and a value of 1 if it was above −5.15; values between −6.85 and −5.15 were linearly interpolated between 0 and 1. Finally, for Score 4, we combined Score 3 with the overall drug-likeness score. The score from the RNN-based drug-likeness model was scaled down by dividing by 100 to ensure values ranged from a minimum of 0 to a maximum of 1, and then we used the average of these scaled values.

### 4.6. Molecular Simulations

The X-ray crystallographic structure of human PARP1, retrieved from the Protein Data Bank (PDB ID: 7KK4) [54], was used for our analysis. Missing side chains within the PARP1 structure were addressed using the conformation prediction tool in Prime version 5.3 [55], followed by the addition of hydrogen atoms at a pH of 7.0. The coordinates of these hydrogen atoms were refined using PROPKA3 [56] to ensure precise placement and overall structural integrity. Restrained energy minimization using the OPLS3 force field was performed to converge within a root-mean-square deviation (RMSD) of less than 0.3 Å [57].

Molecular docking simulations were conducted using Glide-XP in Prime [58,59], focusing on PARP1 protein-ligand interactions. The top-ranking binding pose from compound **1** was integrated into orthorhombic simulation boxes. The box contained 11,969 TIP3P water molecules arranged with a 10 Å buffer distance around the protein structures. To replicate physiological ionic strength (0.15 M) and ensure charge neutrality, 33 Na^+^, and 34 Cl^−^ ions were added to the respective systems.

Molecular dynamics simulations were executed with Desmond version 5.5 [60], utilizing the OPLS3 force field within the NPT ensemble to maintain a fixed number of particles, pressure, and temperature. The particle-mesh Ewald method was used to calculate both long-range and short-range interactions, with a cutoff of 9 Å for van der Waals and electrostatic forces [61]. Temperature was maintained at 300 K using Nose-Hoover thermostats [62], pressure was maintained at 1.01325 bar using Martina-Tobias-Klein barostats [63], and the RESPA integrator was employed to merge the equations of motion, applying a time step of 2.0 femtoseconds for all interactions [64]. A 100 ns simulation was executed following the standard Desmond protocol, capturing conformations and energy data at intervals of 100 and 1.2 picoseconds, respectively. For our analysis, conformations from the 10 ns to 90 ns interval were specifically scrutinized to exclude any initial and terminal phase instabilities.

The quantum mechanical fragment molecular orbital (FMO) method was employed to investigate the interaction between PARP1 and the suggested inhibitors. This approach involved fragmenting each residue of PARP1 at the Cα site and its ligand into distinct fragments, reducing computational costs and mitigating errors. Each fragment’s molecular orbitals were optimized within the system’s electrostatic field using the self-consistent field (SCF) theory [65]. SCF cycles were then performed for fragment pairs (dimers), with energies influenced by the electrostatic field of all other fragments. The energies of monomers and dimers were further analyzed through pair interaction energy decomposition analysis (PIEDA) [66], which broke down interaction energies into electrostatic, exchange repulsion, charge transfer, and dispersion components, providing insights into specific interactions like hydrogen bonds, hydrophobic effects, and steric repulsion. All ab initio FMO calculations were conducted using the General Atomic and Molecular Electronic Structure System (GAMESS) software version 2022 R2 [67], with energy decomposition analysis performed at the resolution of the identity second-order Møller-Plesset perturbation theory (RI-MP2) [68] level, incorporating solvent effects via the polarizable continuum model (PCM) [69] with the 6-31G** basis set (FMO2-RIMP2/PCM/6-31G**). The PCM calculations utilized a one-body PCM (PCM[1]) and a conductor-like PCM (C-PCM) with iterative solvers, while SCF calculations applied a cutoff option ‘RESDIM = 2.0’ to enhance computational efficiency.

## Figures and Tables

**Figure 1 ijms-26-00224-f001:**
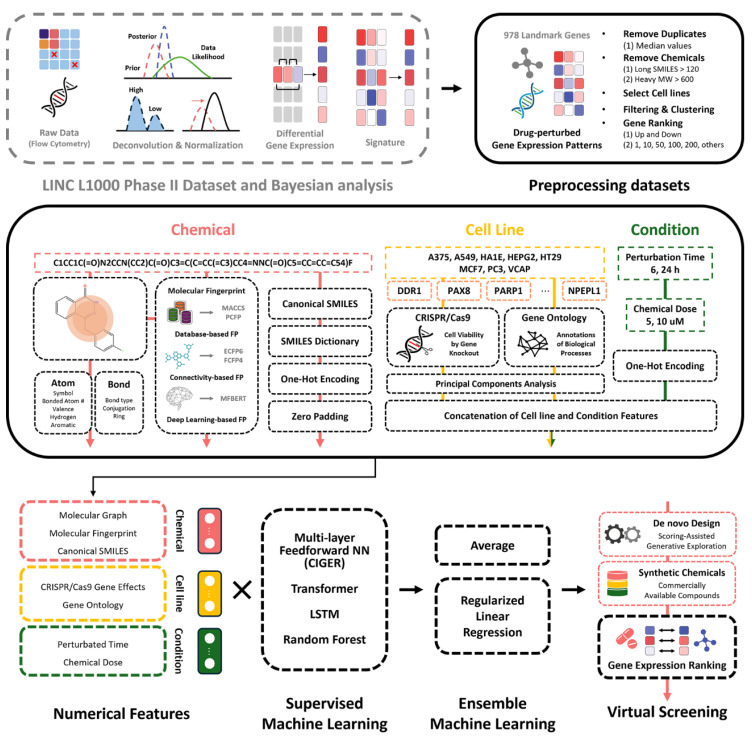
Drug-Induced Gene Expression Ranking Analysis (DIGERA).

**Figure 2 ijms-26-00224-f002:**
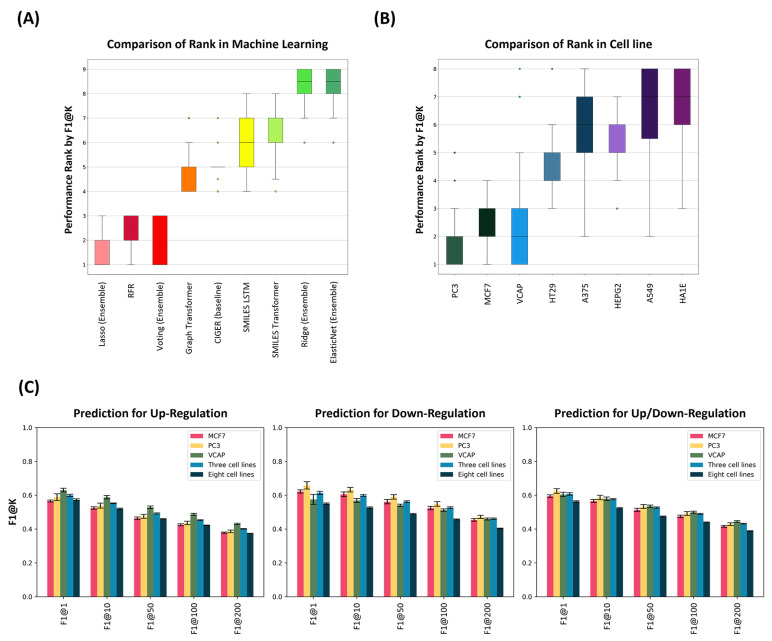
Comparative Analysis of Gene Expression Ranking Prediction Models. (**A**) A boxplot illustrates the comparative rankings of different machine learning models based on the F1@K metrics in the cross-validation dataset. (**B**) A boxplot shows the ranking comparison across various cell lines using the F1@K metrics in the cross-validation dataset. (**C**) Bar plots present the performance of gene expression ranking prediction models trained on different cell line datasets, including MCF7, PC3, VCAP, a combination of three cell lines, and all eight cell lines. The models are evaluated using the F1@K metrics in the cross-validation dataset for predicting up-regulated, down-regulated, and up/down-regulated genes.

**Figure 3 ijms-26-00224-f003:**
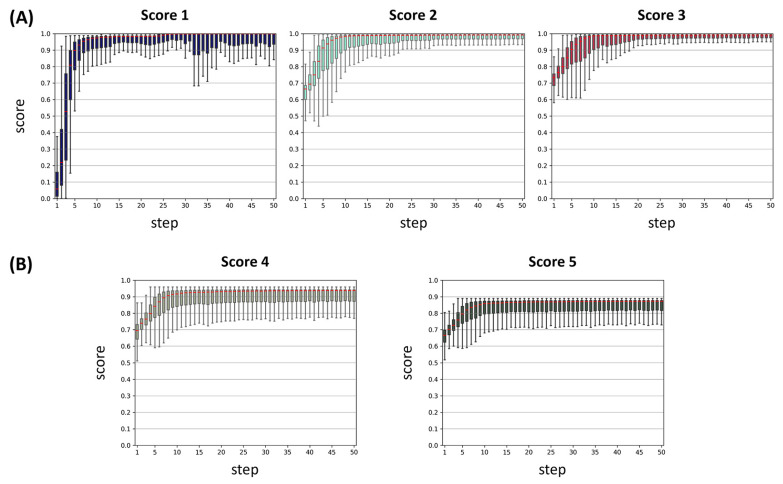
SAGE-based Target Specificity Optimization and Virtual Screening Results for PARP1. (**A**,**B**) Boxplots represent each step in the de novo design of PARP1 inhibitors through the SAGE process: (**A**) Scores 1, 2, and 3; (**B**) Scores 4 and 5. The median of each boxplot is highlighted in red.

**Figure 4 ijms-26-00224-f004:**
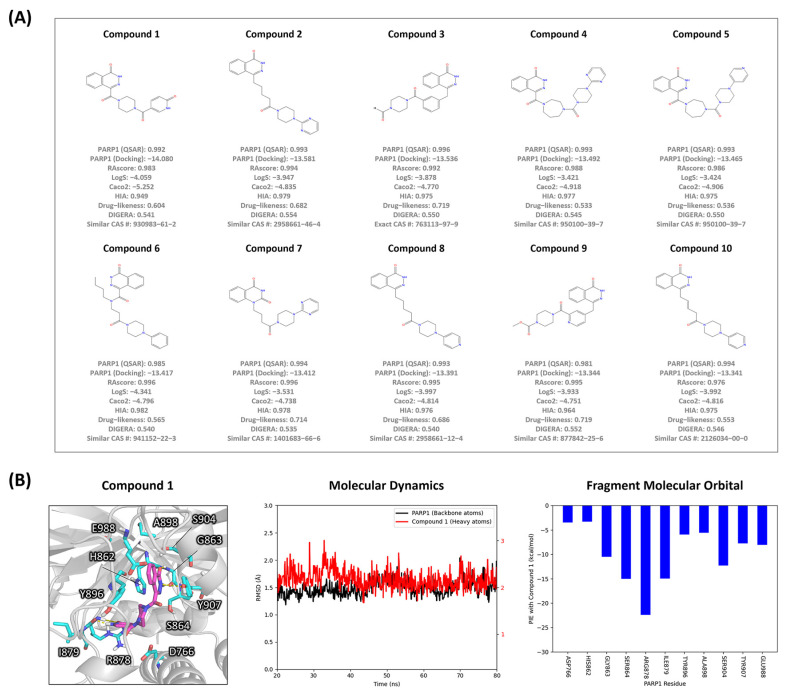
Visualization and Interaction Analysis of Top-ranked Compounds for PARP1. (**A**) The top 10 compounds identified by the SAGE process are illustrated, along with their predicted values. (**B**) The binding mode of compound **1** is illustrated, highlighting key residues of PARP1 identified through quantum-mechanical FMO analysis. The molecular dynamics simulation trajectories of the compound with PARP1 are shown, with the backbone atoms of PARP1 depicted in black and the heavy atoms of the ligand in red. Quantum-mechanical FMO analysis of the complex is presented, focusing on significant interactions that are more stable than −3.0 kcal/mol.

**Table 1 ijms-26-00224-t001:** Summary of Datasets used in this work.

Class	Name	All Set	Training Set	Test Set
Gene Expression Pattern	A375	1888	1510	378
	A549	2136	1709	427
	HA1E	2313	1850	463
	HEPG2	1816	1453	363
	HT29	1873	1498	375
	MCF7	1860	1488	372
	PC3	1709	1367	342
	VCAP	2197	1757	440
PARP1	Active	2808	2241	567
	Inactive	709	567	142
	Decoy	29,863	23,890	5973

**Table 2 ijms-26-00224-t002:** Performance Metrics of Up/Down-Regulated Gene Expression Ranking Prediction in a Single Cell Line.

Class	Cell line	Model	F1@1	F1@10	F1@50	F1@100	F1@200
Up/Down	MCF7	CIGER (baseline)	0.5801 ± 0.0107	0.5523 ± 0.0080	0.5035 ± 0.0081	0.4662 ± 0.0069	0.4095 ± 0.0050
		RFR	0.5917 ± 0.0035	0.5648 ± 0.0095	0.5116 ± 0.0098	0.4732 ± 0.0075	0.4147 ± 0.0057
		Graph Transformer	0.5872 ± 0.0129	0.5546 ± 0.0089	0.5030 ± 0.0087	0.4658 ± 0.0070	0.4093 ± 0.0051
		SMILES Transformer	0.5699 ± 0.0109	0.5433 ± 0.0110	0.4960 ± 0.0096	0.4615 ± 0.0078	0.4070 ± 0.0060
		SMILES LSTM	0.5688 ± 0.0150	0.5434 ± 0.0110	0.4962 ± 0.0101	0.4613 ± 0.0080	0.4070 ± 0.0058
		Voting (Ensemble)	0.6040 ± 0.0100	0.5681 ± 0.0084	0.5173 ± 0.0096	0.4789 ± 0.0077	0.4195 ± 0.0055
		Lasso (Ensemble)	0.5945 ± 0.0074	0.5658 ± 0.0091	0.5131 ± 0.0096	0.4748 ± 0.0075	0.4160 ± 0.0057
		Ridge (Ensemble)	0.5149 ± 0.0822	0.4895 ± 0.0769	0.4484 ± 0.0662	0.4201 ± 0.0561	0.3713 ± 0.0435
		ElasticNet (Ensemble)	0.4342 ± 0.1838	0.4205 ± 0.1641	0.3825 ± 0.1499	0.3632 ± 0.1267	0.3243 ± 0.1028
	PC3	CIGER (baseline)	0.6009 ± 0.0171	0.5713 ± 0.0118	0.5210 ± 0.0105	0.4821 ± 0.0102	0.4215 ± 0.0077
		RFR	0.6283 ± 0.0142	0.5867 ± 0.0150	0.5331 ± 0.0123	0.4915 ± 0.0117	0.4283 ± 0.0086
		Graph Transformer	0.6123 ± 0.0084	0.5750 ± 0.0088	0.5218 ± 0.0099	0.4822 ± 0.0094	0.4217 ± 0.0073
		SMILES Transformer	0.5976 ± 0.0157	0.5641 ± 0.0125	0.5174 ± 0.0104	0.4793 ± 0.0109	0.4208 ± 0.0079
		SMILES LSTM	0.5893 ± 0.0093	0.5591 ± 0.0124	0.5173 ± 0.0106	0.4796 ± 0.0117	0.4205 ± 0.0085
		Voting (Ensemble)	0.6322 ± 0.0140	0.5893 ± 0.0110	0.5379 ± 0.0110	0.4969 ± 0.0102	0.4330 ± 0.0077
		Lasso (Ensemble)	0.6247 ± 0.0132	0.5861 ± 0.0129	0.5328 ± 0.0114	0.4918 ± 0.0111	0.4289 ± 0.0082
		Ridge (Ensemble)	0.3696 ± 0.2405	0.3467 ± 0.2248	0.3246 ± 0.2055	0.3057 ± 0.1812	0.2778 ± 0.1433
		ElasticNet (Ensemble)	0.4639 ± 0.2049	0.4570 ± 0.1575	0.4147 ± 0.1380	0.3905 ± 0.1133	0.3481 ± 0.0957
	VCAP	CIGER (baseline)	0.5698 ± 0.0099	0.5549 ± 0.0077	0.5159 ± 0.0068	0.4846 ± 0.0060	0.4349 ± 0.0046
		RFR	0.6051 ± 0.0143	0.5786 ± 0.0108	0.5308 ± 0.0068	0.4954 ± 0.0062	0.4414 ± 0.0047
		Graph Transformer	0.5739 ± 0.0051	0.5573 ± 0.0070	0.5159 ± 0.0060	0.4847 ± 0.0051	0.4350 ± 0.0038
		SMILES Transformer	0.5418 ± 0.0095	0.5357 ± 0.0093	0.5057 ± 0.0062	0.4790 ± 0.0051	0.4323 ± 0.0044
		SMILES LSTM	0.5477 ± 0.0128	0.5333 ± 0.0096	0.5058 ± 0.0064	0.4784 ± 0.0062	0.4321 ± 0.0051
		Voting (Ensemble)	0.5916 ± 0.0116	0.5759 ± 0.0093	0.5343 ± 0.0075	0.5014 ± 0.0062	0.4479 ± 0.0048
		Lasso (Ensemble)	0.6041 ± 0.0132	0.5791 ± 0.0105	0.5344 ± 0.0073	0.4994 ± 0.0066	0.4447 ± 0.0049
		Ridge (Ensemble)	0.2718 ± 0.2378	0.2651 ± 0.2230	0.2567 ± 0.2058	0.2511 ± 0.1880	0.2354 ± 0.1565
		ElasticNet (Ensemble)	0.3911 ± 0.2237	0.3851 ± 0.2165	0.3696 ± 0.1959	0.3520 ± 0.1795	0.3238 ± 0.1538

**Table 3 ijms-26-00224-t003:** Performance Metrics of Up/Down-Regulated Gene Expression Ranking Prediction in Multiple Cell Lines.

Class	Cell Line	Model	F1@1	F1@10	F1@50	F1@100	F1@200
Up/Down	Three Cell Lines (MCF7, PC3, VCAP)	CIGER (baseline)	0.5904 ± 0.0075	0.5637 ± 0.0066	0.5153 ± 0.0054	0.4792 ± 0.0046	0.4233 ± 0.0033
		RFR	0.6060 ± 0.0083	0.5751 ± 0.0038	0.5249 ± 0.0037	0.4872 ± 0.0032	0.4291 ± 0.0023
		Graph Transformer	0.5915 ± 0.0116	0.5644 ± 0.0051	0.5157 ± 0.0050	0.4794 ± 0.0046	0.4237 ± 0.0033
		SMILES Transformer	0.5724 ± 0.0091	0.5495 ± 0.0060	0.5084 ± 0.0055	0.4752 ± 0.0045	0.4218 ± 0.0034
		SMILES LSTM	0.5714 ± 0.0097	0.5490 ± 0.0056	0.5080 ± 0.0049	0.4749 ± 0.0042	0.4213 ± 0.0034
		Voting (Ensemble)	0.6066 ± 0.0082	0.5776 ± 0.0033	0.5301 ± 0.0038	0.4930 ± 0.0034	0.4343 ± 0.0023
		Lasso (Ensemble)	0.6074 ± 0.0070	0.5762 ± 0.0030	0.5268 ± 0.0042	0.4894 ± 0.0037	0.4310 ± 0.0025
		Ridge (Ensemble)	0.2059 ± 0.2052	0.1970 ± 0.1895	0.1908 ± 0.1706	0.1829 ± 0.1495	0.1809 ± 0.1256
		ElasticNet (Ensemble)	0.3414 ± 0.2387	0.3307 ± 0.2286	0.3152 ± 0.2092	0.2990 ± 0.1886	0.2713 ± 0.1548
	Eight Cell Lines	CIGER (baseline)	0.5317 ± 0.0056	0.4991 ± 0.0041	0.4527 ± 0.0030	0.4210 ± 0.0026	0.3747 ± 0.0018
		RFR	0.5506 ± 0.0033	0.5145 ± 0.0029	0.4673 ± 0.0020	0.4340 ± 0.0019	0.3855 ± 0.0013
		Graph Transformer	0.5351 ± 0.0044	0.5019 ± 0.0035	0.4549 ± 0.0031	0.4227 ± 0.0025	0.3762 ± 0.0014
		SMILES Transformer	0.4950 ± 0.0047	0.4744 ± 0.0033	0.4386 ± 0.0023	0.4112 ± 0.0019	0.3690 ± 0.0016
		SMILES LSTM	0.4931 ± 0.0062	0.4710 ± 0.0043	0.4374 ± 0.0030	0.4103 ± 0.0027	0.3682 ± 0.0018
		Voting (Ensemble)	0.5628 ± 0.0032	0.5264 ± 0.0036	0.4781 ± 0.0026	0.4437 ± 0.0023	0.3930 ± 0.0014
		Lasso (Ensemble)	0.5621 ± 0.0044	0.5233 ± 0.0041	0.4738 ± 0.0025	0.4392 ± 0.0021	0.3886 ± 0.0014
		Ridge (Ensemble)	0.3840 ± 0.1863	0.3693 ± 0.1696	0.3443 ± 0.1471	0.3238 ± 0.1318	0.2923 ± 0.1081
		ElasticNet (Ensemble)	0.3812 ± 0.1735	0.3636 ± 0.1624	0.3383 ± 0.1424	0.3185 ± 0.1273	0.2871 ± 0.1038

**Table 4 ijms-26-00224-t004:** Performance Metrics of QSAR Models for PARP1.

Fingerprints	Model	Train F1-Score	Validation F1-Score	Test F1-Score	Test MCC
MACCS/ECFP6	LGBM	0.974 ± 0.009	0.937 ± 0.120	0.956	0.952
	RFC	0.944 ± 0.017	0.933 ± 0.116	0.938	0.932
	XGB	1.000 ± 0.000	0.921 ± 0.117	0.950	0.945
MACCS/FCFP4	LGBM	0.961 ± 0.013	0.931 ± 0.123	0.948	0.944
	RFC	0.979 ± 0.008	0.924 ± 0.121	0.955	0.951
	XGB	0.997 ± 0.002	0.917 ± 0.124	0.952	0.947
MACCS/MFBERT	LGBM	1.000 ± 0.000	0.913 ± 0.124	0.945	0.940
	RFC	0.949 ± 0.012	0.887 ± 0.105	0.912	0.903
	XGB	1.000 ± 0.000	0.894 ± 0.125	0.937	0.931
MACCS/PCFP	LGBM	0.963 ± 0.013	0.928 ± 0.125	0.940	0.936
	RFC	0.977 ± 0.008	0.913 ± 0.121	0.944	0.939
	XGB	0.999 ± 0.000	0.913 ± 0.126	0.949	0.944

## Data Availability

All results in this work can be found in the Github repository (github.com/hclim0213/DIGERA).

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
