# Peer review of "Development of Drug-Induced Gene Expression Ranking Analysis (DIGERA) and Its Application to Virtual Screening for Poly (ADP-Ribose) Polymerase 1 Inhibitor"

_ijms, 2024, doi:10.3390/ijms26010224_

Round 1
Reviewer 1 Report
Comments and Suggestions for Authors
This study introduces DIGERA (Drug-Induced Gene Expression Ranking Analysis), a Lasso-based ensemble framework designed to predict drug-induced gene expression rankings using LINCS L1000 data. Although the results highlight DIGERA's ability to enhance virtual screening and aid in discovering new PARP1 inhibitors, there was still an issue to be improved.
1. While the paper introduces DIGERA as a novel framework, its reliance on ensemble learning and machine learning methods like Lasso, Random Forest, and GCN builds on established techniques. The novelty primarily lies in integrating these with LINCS L1000 data, but this may be seen as an incremental improvement rather than a groundbreaking innovation. The manuscript should introduce a more unique methodology, such as incorporating multi-omics data or real-time feedback mechanisms in drug design and provide clear examples of how DIGERA addresses unmet needs in drug discovery beyond existing tools.
2. Any reasonable model needs to be verified by experiments. The manuscript should conduct wet-lab experiments to validate at least a subset of the proposed PARP1 inhibitors.
3. The core contents of the introduction have not been clearly presented. For example, 1) what is the existing question? 2) what is your solution? 3) what is the expected outcome, and what is the potential impact of this study? The introduction section of this article is surprisingly long.
Author Response
Response to reviewers’ comments:
I appreciate the reviewers for his/her comments and kindness. My responses and clarifications to each point raised are provided below in red. It’s my pleasure to listen to the reviewers’ comments and learn more.
- Reviewer #1
This study introduces DIGERA (Drug-Induced Gene Expression Ranking Analysis), a Lasso-based ensemble framework designed to predict drug-induced gene expression rankings using LINCS L1000 data. Although the results highlight DIGERA's ability to enhance virtual screening and aid in discovering new PARP1 inhibitors, there was still an issue to be improved.
Comment 1: While the paper introduces DIGERA as a novel framework, its reliance on ensemble learning and machine learning methods like Lasso, Random Forest, and GCN builds on established techniques. The novelty primarily lies in integrating these with LINCS L1000 data, but this may be seen as an incremental improvement rather than a groundbreaking innovation. The manuscript should introduce a more unique methodology, such as incorporating multi-omics data or real-time feedback mechanisms in drug design and provide clear examples of how DIGERA addresses unmet needs in drug discovery beyond existing tools.
Response 1: We sincerely appreciate the reviewer’s thoughtful and constructive comments. While LINCS L1000 data provides comprehensive information on gene expression profiling in cells treated with perturbagens, accurately predicting gene expression profiles remains a significant challenge. As the reviewer suggested, incorporating multi-omics data or real-time feedback mechanisms into model training is indeed a valuable idea. However, such information is not available within the LINCS L1000 dataset. Instead, we maximized the utility of the available data by integrating features such as chemical properties, cell line characteristics, and experimental conditions during training. Notably, our use of CRISPR/Cas9 and Gene Ontology information specific to cell lines distinguishes our approach from existing tools. This innovation enabled DIGERA to outperform recently published models, such as the CIGER model. Furthermore, we have expanded the discussion to highlight the applications of DIGERA, providing additional details in lines 434–442 on page 13 and lines 490-499 on page 14.
Comment 2: Any reasonable model needs to be verified by experiments. The manuscript should conduct wet-lab experiments to validate at least a subset of the proposed PARP1 inhibitors.
Response 2: We sincerely thank the reviewer for this valuable comment and fully acknowledge the importance of experimental validation. However, the primary scope of this study is to establish and demonstrate the computational framework of DIGERA. To address the reviewer’s concern, we conducted a literature search for analogs of the nine novel molecules suggested by DIGERA. Interestingly, six of these analogs were associated with references indicating PARP1 inhibition activity, suggesting that the scaffolds of the proposed molecules are indeed relevant to PARP1 inhibition. Nevertheless, we recognize the necessity of experimental validation for the suggested molecules. To address this limitation, we have expanded the discussion in the manuscript to explicitly outline this point, as described in lines 440-442 on page 13.
Comment 3: The core contents of the introduction have not been clearly presented. For example, 1) what is the existing question? 2) what is your solution? 3) what is the expected outcome, and what is the potential impact of this study? The introduction section of this article is surprisingly long.
Response 3: Thank you for pointing out this important issue. Following the reviewer’s suggestion, we have revised the introduction to clearly articulate the core elements of the study. Specifically, we have clarified: (1) the existing challenge in predicting drug-induced gene expression rankings, (2) how DIGERA addresses this challenge through a Lasso-based ensemble framework leveraging LINCS L1000 data, and (3) the expected outcomes and potential impact of this study, particularly in enhancing virtual drug screening and identifying novel PARP1 inhibitors. Additionally, we have streamlined the introduction to improve clarity and conciseness, as reflected in the revised text in lines 45-51 and 90-91 on page 2.

Reviewer 2 Report
Comments and Suggestions for Authors
This study introduces DIGERA, a Lasso-based integrated learning framework for drug-induced gene expression ranking analysis. By utilizing the LINCS L1000 dataset to predict the ranking of drug-induced gene expression and incorporating numerical features such as molecular graphs, molecular fingerprints, and SMILES, DIGERA demonstrates higher predictive accuracy in gene expression ranking across eight key cell lines compared to baseline models. Then, DIGERA is successfully integrated with Score-based Generative Exploration (SAGE) for virtual screening of PARP1 inhibitors. The manuscript was prepared in adherence to the conventions of scholarly writing and is suitable for acceptance for publication.
Suggestions:
- The paper lacks an in-depth discussion on the limitations of the DIGERA model and potential directions for future enhancements of prediction performance and broadening its scope of applications. The predictive capability of the DIGERA model is constrained by data quality issues inherent in the L1000 dataset, characterized by noise and missing values, as indicated by variations in the predictive performance of the model across various cell lines.
- By merging DIGERA with SAGE, the author carried out virtual screening for PARP1 inhibitors. While all assessments supported the efficacy of this strategy, the proposed compounds still require further experimental validation.
- It would be beneficial to include subheadings in the Results section for better description and clarity.
- Lines 136-137、143、252、532: “The molecular graph captures the three-dimensional connectivity of atoms within a molecule via a graph convolutional network.” In fact, the molecular graph is a two-dimensional representation that utilizes a graph convolutional network to depict the interatomic connections within a molecule. Without the inclusion of additional three-dimensional coordinate information, it can not to directly represent the three-dimensional characteristics of molecules.
- Please review and simplify the description of the tabular data. In line 199: “The Lasso-based ensemble model closely followed with F1-scores...”, as shown in Table 2, the results for the PC3 cell line demonstrate that the RFR model outperforms the Lasso-based ensemble model in terms of F1@1, F1@10, and F1@50. In lines 300 and 334: “... surpassed VCAP in longer-ranking genes”, it seems that the performance of the three-cell-line model on longer-ranking genes does not exceed that of VCAP, as VCAP demonstrates a value of 0.4479 on F1@200 using a voting ensemble model. In line 338: “Therefore, we chose the Lasso-based ensemble model trained...”, the Lasso ensemble model demonstrates superior performance solely in F1@1, whereas the Voting ensemble model excels across a broader spectrum of rankings (Lines 282-286).
- Line 410: “... with eight common residues”: Could the authors clarify why there are 11 residues shown after stating "with eight common residues"?
- It appears that there is a repetition in paragraphs 522-529 and 530-540.
Miner Points:
1. Line 61: Woo et al → Woo et al.
2. Line 63: graph networks → graph neural network
3. Line 71: 1060 → 1060
4. Line 74: SMILES → Simplified Molecular-Input Line-Entry System (SMILES)
5. Line 145: specifics → characteristics
6. Line 165: numerical fingerprint → numerical features
7. Line 359: The accuracy of accuracy → The accuracy
8. Line 379: 0.90 → above 0.90
9. Line 438: value → values
10. Line 470: amount → quantity
11. Line 506: at identical conditions → under identical conditions
12. Line 733: Matthews coefficient → Matthews correlation coefficient
13. In Table 4, "MACCS/ECFP6" and "MACCS/FCFP4" should be consistent with the description in the materials and methods as "MACCS+ECFP6" and "MACCS+FCFP4".
14. Line 357, please provide specific clarification on the diameter of ECFP, specifying whether ECFP4 or ECFP6 is under consideration.
15. It is necessary to include the calculation formula and abbreviated form for the F1-score@K in line 691 and adhere to the expression (such as F1@1, F1@10, or F1-score@1, F1-score@10) in other associated sections.
Author Response
Response to reviewers’ comments:
I appreciate the reviewers for his/her comments and kindness. My responses and clarifications to each point raised are provided below in red. It’s my pleasure to listen to the reviewers’ comments and learn more.
- Reviewer #2
This study introduces DIGERA, a Lasso-based integrated learning framework for drug-induced gene expression ranking analysis. By utilizing the LINCS L1000 dataset to predict the ranking of drug-induced gene expression and incorporating numerical features such as molecular graphs, molecular fingerprints, and SMILES, DIGERA demonstrates higher predictive accuracy in gene expression ranking across eight key cell lines compared to baseline models. Then, DIGERA is successfully integrated with Score-based Generative Exploration (SAGE) for virtual screening of PARP1 inhibitors. The manuscript was prepared in adherence to the conventions of scholarly writing and is suitable for acceptance for publication.
Comment 1: The paper lacks an in-depth discussion on the limitations of the DIGERA model and potential directions for future enhancements of prediction performance and broadening its scope of applications. The predictive capability of the DIGERA model is constrained by data quality issues inherent in the L1000 dataset, characterized by noise and missing values, as indicated by variations in the predictive performance of the model across various cell lines.
Response 1: We appreciate the reviewer’s thoughtful comments and valuable suggestions. We recognize that the predictive capability of DIGERA can be constrained by the inherent data quality issues in the LINCS L1000 dataset, including noise and missing values. As the reviewer suggested, these issues may have contributed to the observed variations in predictive performance across different cell lines. The limitations of the DIGERA model and future direction for improving its predictive performance and expanding its applications have been further discussed in the revised manuscript in lines 434–442 on page 13 and lines 490-499 on page 14.
Comment 2: By merging DIGERA with SAGE, the author carried out virtual screening for PARP1 inhibitors. While all assessments supported the efficacy of this strategy, the proposed compounds still require further experimental validation.
Response 2: Thank you. We sincerely thank the reviewer for this valuable comment and fully acknowledge the importance of experimental validation. However, the primary scope of this study is to establish and demonstrate the computational framework of DIGERA. To address the reviewer’s concern, we conducted a literature search for analogs of the nine novel molecules suggested by DIGERA. Interestingly, six of these analogs were associated with references indicating PARP1 inhibition activity, suggesting that the scaffolds of the proposed molecules are indeed relevant to PARP1 inhibition. Nevertheless, we recognize the necessity of experimental validation for the suggested molecules. To address this limitation, we have expanded the discussion in the manuscript to explicitly outline this point, as described in lines 440-442 on page 13.
Comment 3: It would be beneficial to include subheadings in the Results section for better description and clarity.
Response 3: We appreciate the reviewer’s suggestion to improve the clarity of the Results section. In response, we have added appropriate subheadings to the Results section to better organize the content and enhance readability.
Comment 4: Lines 136-137、143、252、532: “The molecular graph captures the three-dimensional connectivity of atoms within a molecule via a graph convolutional network.” In fact, the molecular graph is a two-dimensional representation that utilizes a graph convolutional network to depict the interatomic connections within a molecule. Without the inclusion of additional three-dimensional coordinate information, it can not to directly represent the three-dimensional characteristics of molecules.
Response 4: We appreciate the reviewer’s insightful comments regarding the dimensional representation of the molecular graph. We agree that the molecular graph primarily represents two-dimensional interatomic connectivity, and we have revised the manuscript accordingly in lines 132, 139, 250, and 537 to clarify this point.
Comment 5: Please review and simplify the description of the tabular data. In line 199: “The Lasso-based ensemble model closely followed with F1-scores...”, as shown in Table 2, the results for the PC3 cell line demonstrate that the RFR model outperforms the Lasso-based ensemble model in terms of F1@1, F1@10, and F1@50. In lines 300 and 334: “... surpassed VCAP in longer-ranking genes”, it seems that the performance of the three-cell-line model on longer-ranking genes does not exceed that of VCAP, as VCAP demonstrates a value of 0.4479 on F1@200 using a voting ensemble model. In line 338: “Therefore, we chose the Lasso-based ensemble model trained...”, the Lasso ensemble model demonstrates superior performance solely in F1@1, whereas the Voting ensemble model excels across a broader spectrum of rankings (Lines 282-286).
Response 5: We appreciate the reviewer’s detailed observations regarding the description and performance comparison of the models. As noted, the results for the PC3 cell line, presented in Table 2, indicate that the Voting ensemble model outperforms the Lasso-based ensemble model in terms of F1@10, F1@50, and F1@200. Similarly, for longer-ranking genes in the three-cell-line model, we acknowledge that the VCAP model achieves a superior F1@200 value of 0.4479. However, as illustrated in Figure 2A, we compared the average performance ranks between single-cell-line models and multi-cell-line models. Although the differences are subtle, the Lasso-based ensemble model demonstrated a slightly better average performance compared to the Voting-based ensemble model, which led us to prioritize the Lasso-based approach. To further clarify the tabular data and provide additional insights, we have visualized the gene expression ranking prediction results as heatmaps. These visualizations have been added to the Supporting Information (Figures S1–S3).
Comment 6: Line 410: “... with eight common residues”: Could the authors clarify why there are 11 residues shown after stating "with eight common residues"?
Response 6: We revised the manuscript in line 411 on page 12.
Comment 7: It appears that there is a repetition in paragraphs 522-529 and 530-540.
Response 7: We revised the manuscript by removing the repeated paragraph.
Comment 8: Line 61: Woo et al → Woo et al.
Response 8: We revised the manuscript in line 55 on page 2.
Comment 9: Line 63: graph networks → graph neural network
Response 9: We revised the manuscript in line 57 on page 2.
Comment 10: Line 71: 1060 → 1060
Response 10: We revised the manuscript in line 65 on page 2.
Comment 11: Line 74: SMILES → Simplified Molecular-Input Line-Entry System (SMILES)
Response 11: We revised the manuscript in line 68 on page 2.
Comment 12: Line 145: specifics → characteristics
Response 12: We revised the manuscript in line 141 on page 4.
Comment 13: Line 165: numerical fingerprint → numerical features
Response 13: We revised the manuscript in line 161 on page 5.
Comment 14: Line 359: The accuracy of accuracy → The accuracy
Response 14: We revised the manuscript in line 359 on page 10.
Comment 15: Line 379: 0.90 → above 0.90
Response 15: We revised the manuscript in line 381 on page 11.
Comment 16: Line 438: value → values
Response 16: We revised the manuscript in line 443 on page 13.
Comment 17: Line 470: amount → quantity
Response 17: We revised the manuscript in line 475 on page 14.
Comment 18: Line 506: at identical conditions → under identical conditions
Response 18: We revised the manuscript in line 519 on page 15.
Comment 19: Line 733: Matthews coefficient → Matthews correlation coefficient
Response 19: We revised the manuscript in line 738 on page 19.
Comment 20: In Table 4, "MACCS/ECFP6" and "MACCS/FCFP4" should be consistent with the description in the materials and methods as "MACCS+ECFP6" and "MACCS+FCFP4".
Response 20: We revised the manuscript in lines 733-734 on page 19.
Comment 21: Line 357, please provide specific clarification on the diameter of ECFP, specifying whether ECFP4 or ECFP6 is under consideration.
Response 21: We revised the manuscript in line 357 on page 10.
Comment 22: It is necessary to include the calculation formula and abbreviated form for the F1-score@K in line 691 and adhere to the expression (such as F1@1, F1@10, or F1-score@1, F1-score@10) in other associated sections.
Response 22: We appreciate the reviewer’s attention to detail regarding the expression of the F1@K (F1-score@K). To address this comment, we have standardized the notation to F1@K throughout the manuscript, including all relevant figures (Figure 2), tables (Tables 2–3, Tables S3–S7), and associated sections in lines 173, 239, 241, 244, 696, and 697. Additionally, we have included the calculation formula for the F1@K in line 698 on page 18.

Reviewer 3 Report
Comments and Suggestions for Authors
In this study, the authors present a new machine learning approach - Drug-Induced Gene Expression Ranking Analysis (DIGERA).
Using those ML models with features for molecules, cell lines, and experimental conditions, DIGERA showed high performance in ranking gene expression. The authors further did the virtual screening for new PARP1 inhibitors using DIGERA and reported ten candidate compounds. They carried the molecular docking, molecular dynamics simulations, and FMO analysis. All candidate compounds showed strong binding to PARP1.
In summary, the authors describe the ability of DIGERA to aid drug development. They also point out the limitations of their approach, regarding the used datasets and the lack of experimental validation. However, I beleive that this could be a significant in silico approach for further development.
I think that the paper is in a publishable form.
Author Response
Response to reviewers’ comments:
I appreciate the reviewers for his/her comments and kindness. My responses and clarifications to each point raised are provided below in red. It’s my pleasure to listen to the reviewers’ comments and learn more.
- Reviewer #3
In this study, the authors present a new machine learning approach - Drug-Induced Gene Expression Ranking Analysis (DIGERA).
Comment 1: Using those ML models with features for molecules, cell lines, and experimental conditions, DIGERA showed high performance in ranking gene expression. The authors further did the virtual screening for new PARP1 inhibitors using DIGERA and reported ten candidate compounds. They carried the molecular docking, molecular dynamics simulations, and FMO analysis. All candidate compounds showed strong binding to PARP1. In summary, the authors describe the ability of DIGERA to aid drug development. They also point out the limitations of their approach, regarding the used datasets and the lack of experimental validation. However, I beleive that this could be a significant in silico approach for further development. I think that the paper is in a publishable form.
Response 1: Thank you for your positive and encouraging feedback. In this study, we utilized features representing molecules, cell lines, and experimental conditions to maximize the learning potential of the LINCS L1000 data, achieving superior performance compared to existing models. Furthermore, we enhanced model performance through the application of ensemble learning techniques. As an example of DIGERA's application in drug discovery, we suggested potential PARP1 inhibitors, demonstrating the framework's ability to aid in the drug discovery process. We appreciate your acknowledgment of DIGERA's potential as a significant in-silico approach for future development.

Reviewer 4 Report
Comments and Suggestions for Authors
The main question addressed by the research is how to efficiently predict drug-induced gene expression rankings using machine learning, specifically through the development of DIGERA, and its application to virtual screening for Poly (ADP-Ribose) Polymerase 1 (PARP1) inhibitors.
It addresses a significant gap in the scalability and reliability of phenotype-based drug screening methods. While existing methods like DeepCOP or DeepCE have advanced gene expression profiling, DIGERA introduces an ensemble learning framework that incorporates multiple feature representations (e.g., molecular graphs, fingerprints, and SMILES). This multifaceted approach addresses the limitations of experimental noise and missing values in high-throughput datasets, making it a valuable contribution.
Using molecular graphs, SMILES, and ensemble models is a notable improvement over models that rely on single representations.
The incorporation of the SAGE method for virtual screening and de novo design of drug-like molecules is a practical application rarely explored in prior methods.
Comparative analyses indicate that DIGERA outperforms baseline and individual models, demonstrating its robustness across different cell lines and experimental setups.
While the authors preprocess the LINCS L1000 dataset, additional imputation methods or benchmarking against alternative datasets (e.g., CMap) could enhance robustness.
The study relies heavily on computational predictions. At least one experimental validation of a predicted PARP1 inhibitor would strengthen the claims.
Some figures (e.g., Figure 4 and Figure 3) are dense and would benefit from clearer annotations or simplifications to improve readability.
Tables summarizing model performances (e.g., Table 2 and Table 3) are thorough but could be supplemented with visual aids (e.g., heatmaps) for easier interpretation.
The depiction of molecular structures (e.g., in Figure 4) is helpful but could be enhanced by highlighting critical interactions or binding modes visually.
Incorporating additional metrics, such as ROC AUC or precision-recall curves, could provide a more comprehensive evaluation of model performance, particularly for imbalanced datasets.
A brief discussion comparing DIGERA with cutting-edge methods like graph-based deep learning frameworks could further contextualize its novelty.
The conclusions are consistent with the evidence presented. The manuscript demonstrates that DIGERA can predict gene expression rankings effectively and identifies novel PARP1 inhibitors. The analyses and results align with the main research question. However, stronger experimental validation and deeper interpretability analyses would make the conclusions more robust.
Comments on the Quality of English Languagetypo check
Author Response
Response to reviewers’ comments:
I appreciate the reviewers for his/her comments and kindness. My responses and clarifications to each point raised are provided below in red. It’s my pleasure to listen to the reviewers’ comments and learn more.
- Reviewer #4
The main question addressed by the research is how to efficiently predict drug-induced gene expression rankings using machine learning, specifically through the development of DIGERA, and its application to virtual screening for Poly (ADP-Ribose) Polymerase 1 (PARP1) inhibitors. It addresses a significant gap in the scalability and reliability of phenotype-based drug screening methods. While existing methods like DeepCOP or DeepCE have advanced gene expression profiling, DIGERA introduces an ensemble learning framework that incorporates multiple feature representations (e.g., molecular graphs, fingerprints, and SMILES). This multifaceted approach addresses the limitations of experimental noise and missing values in high-throughput datasets, making it a valuable contribution. Using molecular graphs, SMILES, and ensemble models is a notable improvement over models that rely on single representations. The incorporation of the SAGE method for virtual screening and de novo design of drug-like molecules is a practical application rarely explored in prior methods. Comparative analyses indicate that DIGERA outperforms baseline and individual models, demonstrating its robustness across different cell lines and experimental setups.
Comment 1: While the authors preprocess the LINCS L1000 dataset, additional imputation methods or benchmarking against alternative datasets (e.g., CMap) could enhance robustness.
Response 1: We sincerely appreciate the reviewer’s insightful suggestion. As noted, CLUE (CMap and LINCS Unified Environment) provides valuable datasets beyond LINCS L1000, including information on cell viability, chromatin profiling, and protein phosphorylation. While these datasets indeed offer valuable insights, this study specifically focuses on predicting gene expression profiling induced by chemical perturbations in specific cell lines. Incorporating additional datasets would extend beyond the current scope of this work. Regarding gene expression profiling, the LINCS L1000 dataset is the largest available resource. However, it is inherently noisy and subject to biases, as it aggregates data from multiple experimental setups. These challenges have historically resulted in lower performance for existing published models. To address this, our study maximized the use of LINCS L1000 data by leveraging features that represent molecules, cell lines, and experimental conditions. This approach significantly improved performance compared to existing models, and ensemble learning further enhanced the robustness and accuracy of the framework. Moreover, expanding its applications has been further discussed in the revised manuscript in lines 490-499 on page 14.
Comment 2: The study relies heavily on computational predictions. At least one experimental validation of a predicted PARP1 inhibitor would strengthen the claims.
Response 2: We sincerely appreciate the reviewer’s insightful comment and fully acknowledge the importance of experimental validation. However, the scope of this study is focused on computational analysis to establish and demonstrate the DIGERA framework. To address this concern, we performed a literature search on analogs of the nine novel molecules suggested by DIGERA. Notably, six of these analogs were found to have references supporting their relevance to PARP1 inhibition. This suggests that the scaffolds of the proposed molecules are indeed associated with PARP1 inhibition. Nonetheless, we recognize that experimental validation is still necessary to confirm the activity of the suggested molecules. To address this limitation, we have expanded the discussion section to explicitly acknowledge this point, as outlined in lines 438-442 on page 13.
Comment 3: Some figures (e.g., Figure 4 and Figure 3) are dense and would benefit from clearer annotations or simplifications to improve readability.
Response 3: We appreciate the reviewer’s valuable feedback regarding the clarity and readability of the figures. As suggested, we have made several modifications to Figure 3 and Figure 4 to improve their presentation. Specifically, we have increased the font size, adjusted the layout to reduce density, and added clearer annotations to ensure the figures are more readable and accessible. These changes enhance clarity and allow the data to be more easily interpreted by readers.
Comment 4: Tables summarizing model performances (e.g., Table 2 and Table 3) are thorough but could be supplemented with visual aids (e.g., heatmaps) for easier interpretation.
Response 4: Thank you for your valuable suggestion. As recommended by the reviewer, we have created heatmaps to visually supplement the data presented in Tables 2 and 3. These heatmaps have been added to Figures S1-S3 in Supporting Information to enhance interpretability and accessibility.
Comment 5: The depiction of molecular structures (e.g., in Figure 4) is helpful but could be enhanced by highlighting critical interactions or binding modes visually.
Response 5: Thank you for your insightful suggestion. As recommended by the reviewer, we have revised Figure 4 to include clearer annotations, visually highlighting critical interactions and binding modes to enhance clarity and understanding.
Comment 6: Incorporating additional metrics, such as ROC AUC or precision-recall curves, could provide a more comprehensive evaluation of model performance, particularly for imbalanced datasets.
Response 6: We sincerely appreciate the reviewer’s insightful comment regarding the inclusion of additional evaluation metrics. In response, we have incorporated the ROC AUC (AUROC) values for up- and down-regulated gene expression rankings to provide a more comprehensive assessment of DIGERA’s performance. These results have been added to Table S8 in the Supporting Information. The AUROC metric offers a more accurate and meaningful evaluation, particularly in the context of imbalanced datasets, and further demonstrates the robustness of the proposed model.
Comment 7: A brief discussion comparing DIGERA with cutting-edge methods like graph-based deep learning frameworks could further contextualize its novelty.
Response 7: Thank you for this valuable suggestion. We have added a discussion comparing DIGERA with other cutting-edge graph-based deep learning frameworks to better contextualize its novelty. This comparison can be found in the discussion section in lines 434-438 on page 13.
Comment 8: The conclusions are consistent with the evidence presented. The manuscript demonstrates that DIGERA can predict gene expression rankings effectively and identifies novel PARP1 inhibitors. The analyses and results align with the main research question. However, stronger experimental validation and deeper interpretability analyses would make the conclusions more robust.
Response 8: We sincerely appreciate the reviewer’s thoughtful comment. We fully acknowledge the importance of experimental validation. However, the focus of this study was to develop a computational model leveraging LINCS L1000 data to predict gene expression profiling for novel molecules. To provide indirect evidence supporting the predictions, we conducted a literature search on analogs of the nine novel molecules suggested by DIGERA. Notably, six of these analogs were found to have references indicating their relevance to PARP1 inhibition. This suggests that the scaffolds of the proposed molecules are associated with PARP1 inhibition. Nonetheless, we recognize that direct experimental validation remains essential. To address this limitation, we have expanded the discussion section to explicitly outline the constraints of this study, as described in lines 434–442 on page 13 and lines 490-499 on page 14.

Round 2
Reviewer 1 Report
Comments and Suggestions for Authors
The manuscript can be punlished
Reviewer 4 Report
Comments and Suggestions for Authors
The authors have addressed all comments, and I recommend publishing the article.
Comments on the Quality of English Languagen/a